# Effect of Hydrothermal Method Temperature on the Spherical Flowerlike Nanostructures NiCo(OH)_4_-NiO

**DOI:** 10.3390/nano12132276

**Published:** 2022-07-01

**Authors:** Kai Wang, Meini Yuan, Xiaochen Cao, Congming Ding, Jian Ma, Zeyuan Wei

**Affiliations:** College of Mechatronic Engineering, North University of China, Taiyuan 030051, China; wangkai007x@163.com (K.W.); cxc2065216848@163.com (X.C.); dingcm2738@163.com (C.D.); m821207588@163.com (J.M.); weizeyuan1122@163.com (Z.W.)

**Keywords:** hydrothermal deposition, electrophoretic deposition, electrode material, supercapacitor

## Abstract

NiCo(OH)_4_-NiO composite electrode materials were prepared using hydrothermal deposition and electrophoretic deposition. NiCo(OH)_4_ is spherical and flowerlike, composed of nanosheets, and NiO is deposited on the surface of NiCo(OH)_4_ in the form of nanorods. NiCo(OH)_4_ has a large specific surface area and can provide more active sites. Synergistic action with NiO deposits on the surface can provide a higher specific capacitance. In order to study the influence of hydrothermal reaction temperature on the properties of NiCo(OH)_4_, the prepared materials of NiCo(OH)_4_-NiO, the hydrothermal reaction temperatures of 70 °C, 90 °C, 100 °C, and 110 °C were used for comparison. The results showed that the NiCo(OH)_4_-NiO-90 specific capacitance of the prepared electrode material at its maximum when the hydrothermal reaction temperature is 90 °C. The specific capacitance of the NiCo(OH)_4_-NiO-90 reaches 2129 F g^−1^ at the current density of 1 A g^−1^ and remains 84% after 1000 charge–discharge cycles.

## 1. Introduction

With the development of society and the increase in energy consumption, the exploration of efficient and pollution-free energy storage equipment has attracted the attention of researchers [1,2,3,4]. At present, representative energy storage devices include lithium-ion batteries [5], fuel cells [6] and supercapacitors [7,8,9]. Among them, supercapacitors, especially pseudocapacitors, have attracted more and more researchers’ attention due to their high power density, fast charge–discharge capacity, and cycling stability [10,11]. They are expected to become a new generation of energy storage equipment. Supercapacitors’ electrodes use chemical reactive metals, transition metal oxides [12], sulfides [13], and conductive polymers [14]. In the working process of supercapacitors, not only does the adsorption–desorption of ions occur between the electrode material and electrolyte, but the electrode material also participates in Oxidation-Reduction Reactions (REDOX) [15], so as to realize the function of storing and releasing energy, providing a higher potential window, relatively high specific capacitance and a faster rate of charge–discharge. However, due to the REDOX reaction of electrode materials, the service life of supercapacitors decreases with the increase in charge–discharge cycles [16]. Therefore, the service life of supercapacitors is also one of the issues that researchers pay attention to.

When metal and transition metal oxide materials [17], such as Ni and Co [18], are used as electrode materials for supercapacitors, they have the advantages of wide potential window, high capacitance, good conductivity, simple preparation, and low cost, etc., and have great application potential in future constrained capacitor electrode materials [19]. However, compared with multi-carbon materials, metals and transition metal oxide electrode materials have a smaller specific surface area, which results in smaller capacitors and poorer cycle life. Therefore, how to improve the specific surface area and structural stability of pseudocapacitor electrode materials is the focus of current research on electrode materials [20].

In the current research, researchers have focused on improving the specific surface area of transition metals so as to improve the specific capacitance and cycle life. For example, Jiang et al. [21] prepared the metal oxides composite electrode materials NiO/Ni@C, which had a maximum specific capacitance of 805.3 F g^−1^ at the current density of 0.5 A g^−1^, and it had a high-rate performance. The capacity remained at 69.7% after 10,000 charge–discharge cycles. Xiong et al. [22] successfully prepared the metal–organic framework material NiO@Ni-MOF/NF electrode materials. Due to their porous structure, the electrode materials not only had lower impedance, but also a specific capacitance of up to 1853 F g^−1^ at the current density of 1 A g^−1^. The capacity retention rate reached 94% after 3000 charge–discharge cycles. Manibalan et al. [23], prepared NiO@Ni(OH)_2_-Î±-MoO_3_ nanocomposites with nanosheet and nanospherical heterostructures using a hydrothermal method. They showed a high specific capacity of 445 F g^−1^ at the current density of 1 A g^−1^. After 3000 cycles at 10 A g^−1^, the capacitance retention rate reached 97.3%. Wang et al. [24] grew NiO in situ on nickel foam, and the NiO@Ni electrode material prepared thus had an amazing cycle performance. After 1500 cycles of charge–discharge, the specific capacitance was still maintained at 100%. Huang et al. [25] successfully prepared NiCo-LDH@MoO_3_/NF composite electrode material with a double-layer structure via two-step electrophoresis deposition. After testing, the specific capacitance of this material reached 1904 F g^−1^ at the current density of 1 A g^−1^, and the specific capacitance retention rate was 86% after 10,000 charge–discharge cycles at the current density of 20 A g^−1^, showing excellent cycle stability. Schiavi et al. [26] prepared Ni(OH)_2_@NiO core–shell materials using a two-step hydrothermal method combined with electrophoretic deposition. By controlling different etching temperatures, the lengths of the nanowires were different. Finally, it was concluded that with the increase in the reaction temperature, the length of the nanowires increased first and then decreased, and the capacitance value changed with the length of the nanowires, reaching the highest value at 70 °C. Mozaffari et al. [27] prepared NiO@Ni(OH)_2_ nano-array electrode materials through two-step electrophoretic deposition and attached metal hydroxides to the metal oxide core, which not only greatly improved the electrochemical performance, but the specific capacitance was also almost uselessly reduced after 5000 charge–discharge cycles. Using a hydrothermal method combined with electrophoretic deposition, the specific surface area and specific capacitance of nanostructured electrode materials containing metal oxides were greatly improved [20].

In this paper, the electrode materials were prepared using a hydrothermal method combined with electrophoretic deposition. In order to study the influence of hydrothermal temperature on the performance of prepared electrode materials, this paper used Ni and Co as raw materials. Hydrothermal reaction temperatures of 70 °C, 90 °C, 100 °C, and 110 °C were used. After the reaction, the precursors were deposited via electrophoresis. The composition and surface morphology were observed via XRD and SEM, and their electrochemical properties were compared. The optimum performance of the preparation process was calculated, which provided the basis for the actual industrial production. The preparation process is simple and controllable, at the same time the materials are cheap, environmentally friendly, and pollution free.

## 2. Materials and Methods

### 2.1. Preparation of the Precursor, NiCo(OH)_4_

All the chemicals involved in this work were of analytical grade and directly utilized without any further purification. As shown in Figure 1, 0.5 mmol NiCl_2_ · 6H_2_O (all reagents in this papper come from Sinopharm Chemical ReagentCo., Ltd., Shanghai, China) and 1 mmol Co(NO_3_)_2_ · 6H_2_O were dissolved in 70 mL deionized water, stirred by a magnetic agitator, and completely dissolved. 0.5 mmol of urea (CH_4_N_2_O) and 0.5 mmol of NH_4_F were added to the dissolved solution and stirred until the solution was completely dissolved. Then, the above solution was placed in a 50 mL stainless steel reaction kettle lined with polytetrafluoroethylene (PTFE), and nickel foam was added as the reaction attachment. The hydrothermal reaction conditions reacted at 70 °C, 90 °C, 100 °C and 110 °C for 7 h [22]. The obtained precursors were labeled NiCo(OH)_4_-70, NiCo(OH)_4_-90, NiCo(OH)_4_-100 and NiCo(OH)_4_-110, respectively.

### 2.2. Preparation of Nano-Cere-Shell Materials NiCo(OH)_4_-NiO

First, 2 mol Ni(NO_3_)_2_ · 6H_2_O was dissolved in 50 mL deionized water, stirred with a magnetic agitator for half an hour until it completely dissolved, and then the solution was used as electrolyte. NiCo(OH)_4_-NiO nanostructured electrode materials were prepared using nickel foam grown with NiCo(OH)_4_ as the working electrode, and platinum as the counter electrode. The i-t chronoamperometry was used for electrophoretic deposition. The voltage was set at 0.5 V, the sample interval was 1 s, and the scanning time was 300 s [26,28,29]. The spherical flowerlike nanoelectrode materials composed of nanosheets were obtained and labeled NiCo(OH)_4_-NiO-70, NiCo(OH)_4_-NiO-90, NiCo(OH)_4_-NiO-100, and NiCo(OH)_4_-NiO-110, respectively.

### 2.3. Characterizations and Electrochemical Measurements

The morphology of the NiCo(OH)_4_-NiO were directly examined using X-ray photoelectron spectroscopy (XPS, ESCALAB 250Xi, Thermo Scientific Company, Walsham, Massachusetts, USA), X-ray diffraction (XRD, D/max 2600, Rigaku, Japan), scanning electron microscopy (SEM, TESCAN MIRA4, MERLIN Compact, Brno, Czech Republic), and transmission electron microscopy (TEM, Tecnai G2 F20 S-TWIN, Oregon, USA).

### 2.4. Electrochemical Properties Measurements

In 1 M KOH aqueous solution, platinum was used as the counter electrode, HgO was used as the reference electrode, and the above materials were used as the working electrode for electrochemical tests. According to Equation (1), we can calculate the specific capacitance of the electrode material [17].
(1)$$C_{s}=\frac{I∆t}{m∆V}$$
where I is the current density, ∆t is the discharge time, m is the active material mass, and ∆V is the discharge potential window.

## 3. Results and Discussion

### 3.1. Composition

Figure 2 shows the XRD pattern of the nano electrode material NiCo(OH)_4_-NiO after electrophoretic deposition. It can be seen from Figure 2 that the diffraction peaks of Ni(OH)_2_ are 38.6°, 52.1° and 73.0°, respectively, corresponding to the (0 0 2), (0 1 2), and (3 2 1) planes of cubic Ni(OH)_2_ phase. The diffraction peaks of Co(OH)_2_ are 19.1°, 32.6° and 51.5°, respectively, corresponding to the (0 0 1), (1 0 0) and (0 1 2) planes of cubic Co(OH)_2_ phase. The diffraction peaks of NiO are 37.2° and 59.8°, respectively, corresponding to the (1 1 1) and (2 2 0) planes of cubic NiO phase. There are no diffraction peaks except those of Ni(OH)_2_, Co(OH)_2_ and NiO, indicating that C in NiCo(OH)_4_-NiO is amorphous [2,30].

In order to further analyze the surface composition and elemental valence states of the prepared NiCo(OH)_4_-NiO electrode material, XPS tests were performed on the materials. The results of XPS are shown in Figure 3. It can be seen from Figure 3a that the prepared materials are mainly composed of Ni, Co, and O, which is consistent with the XRD measurement results. In Figure 3b, the peak of Ni 2p represented the presence of 2p_3/2_ and 2p_2/1_. The peaks of characteristic peaks are 854.9 eV and 872.5 eV, respectively. Corresponding to Ni 2p_3/2_ and Ni 2p_1/2_, it indicates that NiO is generated on the foam nickel after electrophoretic deposition. The peak values of Ni 2p correspond to 560.9 eV and 578.7 eV, respectively, via a Gaussian fitting method [31], indicating that Ni existed in the form of Ni(OH)_2_. The presence of satellite peaks in Ni 2p orbit can further indicate the existence of NiO [32]. The XPS spectrum of Co is shown in Figure 3c. There is only one orbit, corresponding to a peak of 780 eV, and its weak satellite peak indicates that a small amount of Co_3_O_4_ is generated in the reaction process. The XPS spectrum of O 1s is shown in Figure 3d. There are two clear characteristic peaks that can be seen, corresponding to -OH^−^ (284.8 eV) and NiO (286.1 eV), respectively, indicating the presence of hydroxyl groups in the product. The peak of O_2_ indicates that NiO is present in crystalline form [33,34].

The specific surface area of the electrode material was characterized by N_2_ adsorption–desorption tests, as shown in Figure 4. Figure 4 is the N_2_ adsorption–desorption isotherm of NiCo(OH)_4_-NiO-70. It can be seen from the graph that the N_2_ adsorption–desorption isotherm of NiCo(OH)_4_-NiO-70 is a typical type IV with H3-type hysteresis loop, indicating that there is mesoporous structure in the sample. The volume of specific surface area and pore calculated by Emmet–Teller (BET) is 13 m^2^ g^−^^1^ and 0.052 cm^3^ g^−1^, respectively. Similarly, it can be seen that NiCo(OH)_4_-NiO-90, NiCo(OH)_4_-NiO-100, and NiCo(OH)_4_-NiO-110 also have mesoporous structure. The volume of the specific surface area and pore of NiCo(OH)_4_-NiO-90 is 33 m^2^ g^−1^ and 0.088 cm^3^ g^−1^, respectively; The volume of the specific surface area and pore of NiCo(OH)_4_-NiO-100 is 25 m^2^ g^−1^ and 0.073 cm^3^ g^−1^, respectively; The volume of the specific surface area and pore volume of NiCo(OH)_4_-NiO-110 is 21 m^2^ g^−1^ and 0.075 cm^3^ g^−1^, respectively. Through a N_2_ adsorption–desorption test, it can be concluded that NiCo(OH)_4_-NiO-90 has the largest specific surface area.

### 3.2. Surface Microtopography

Figure 5 shows SEM tests of electrode materials prepared after capacitor deposition at different hydrothermal reaction temperatures. Figure 5a,b show SEM images of NiCo(OH)_4_-NiO-70 electrode materials at different magnifications prepared at hydrothermal reaction temperature of 70 °C, respectively. It can be clearly observed that the surface of the electrode material is composed of irregular layered nanosheets, which covers the surface of the nickel foam in a honeycomb form. When the hydrothermal reaction temperature increased from 70 °C to 90 °C, the thickness of the electrode material microstructure nanosheets became thinner. It can be observed from Figure 5c,d that the nanosheets appeared regularly in combinations [25,27], forming a petal spherical structure, and uniformly covered the nickel foam. Compared with nanocrystalline NiCo(OH)_4_-NiO-70 electrode material, the microstructure of NiCo(OH)_4_-NiO-90 electrode material had a larger surface area, which could provide more active sites; this can be supported by N_2_ adsorption–desorption tests. Theoretically, NiCo(OH)_4_-NiO-90 electrode material improves the electrochemical properties of the material. However, as the hydrothermal reaction temperature continues to rise, when the temperature reached 100 °C, as shown in Figure 5e,f, the distance between the NiCo(OH)_4_-NiO-100 nanosheets became smaller, the size of the petal spherical structure composed of the nanosheets decreased significantly, and the petal spherical structure composed of the nanosheets demonstrated dense stacking. The specific surface area of such microstructures decreases compared with that of NiCo(OH)_4_-NiO-90, and in theory, the specific capacitance value of NiCo(OH)_4_-NiO-100 also decreases. It can be clearly observed in Figure 5g,h that when the hydrothermal reaction temperature rises to 110 °C, the nanosheets present an obvious stacking phenomenon. The space between the nanosheets became smaller, and no obvious petal spherical structure formed. At the same time, the NiCo(OH)_4_-NiO-110 electrode material not only has a small specific surface area, but also decreases the distance between the nanosheets, which affects the free movement of electrons and ions in the charging and discharging process and limits the specific capacitance.

During the hydrothermal process, the following reactions occurred [23,24]:CH_4_N_2_O + H_2_O ↔ NH_3_ + CO_2_(2)
NH_3_ + H_2_O^−^ ↔ NH_4_^+^ + OH^−^(3)
Ni^2+^+ Co^2+^+ 4OH^−^ ↔ NiCo(OH)_4_(4)

At different reaction temperatures, the morphologies of NiCo(OH)_4_-NiO crystals are different, as shown in Figure 5. It can be seen from Figure 5 that, at 70 °C, the nanostructures that formed are flaky and are covered on the nickel foam with the intricate morphology of a honeycomb network. When the temperature rises to 90 °C, the nanosheets begin to combine and interlace with each other, forming a petal spherical microstructure, and the specific surface area of the electrode material increases obviously. This is because when the temperature increases, the hydrothermal reaction becomes more intense and the growth rate of NiCo(OH)_4_-NiO nanosheet crystals is faster, leading to accumulation and combination into spherical structures [35]. When the temperature of water reaches 100 °C, the formation rate of nanosheets is further accelerated. At high temperature, the reaction is more intense, and the pressure increases during the reaction process, resulting in stacking limitation of nanosheets and smaller distance between them [36,37]. When the temperature reaches 110 °C, the agglomeration phenomenon between the nanosheets is more obvious, and even the nanosheets are bonded together instead of the petal ball. In this condition, the specific surface area of the electrode material decreases significantly, and the distance between the nanosheets is at its minimum. Theoretically, among the electrode materials prepared at different hydrothermal temperatures, the NiCo(OH)_4_-NiO-90 electrode material obtained at hydrothermal reaction temperature of 90 °C has the largest specific capacitance and the fastest electron transfer speed.

After electrophoretic deposition, NiO is attached to the surface of NiCo(OH)_4_ as a nanorod-like structure, forming a composite structure with a more cohesive structure [38]. Compared with the nanosheet NiCo(OH)_4_ electrode material before electrophoretic deposition, NiCo(OH)_4_-NiO obviously has a larger specific surface area. The synergistic effect of NiO and NiCo(OH)_4_ not only provides more active sites, but also the electrode material participates in REDOX reactions, resulting in higher specific capacitance [39,40] and better pseudocapacitance performance.

In order to further observe and analyze the microscopic morphology of petal spherical NiCo(OH)_4_-NiO electrode material obtained after electrophoretic deposition, TEM tests were performed on the material, as shown in Figure 6. Figure 6a shows a TEM test of the electrophoretic deposition of NiO on nanocrystal NiCo(OH)_4_ electrode material prepared at hydrothermal reaction temperature of 90 °C. It can be seen from Figure 6a that NiO attaches to NiCo(OH)_4_ with nanocrystal rod structure, forming a more complex structure that can greatly improve the specific surface area of the electrode material. Additionally, it provides more active sites for charge transfer [38]. The synergistic effect of NiCo(OH)_4_ and NiO provides more directionally transferred electrons in the charging and discharging process, and more active substances participate in the REDOX reaction. Figure 6b shows the SAED diffraction rings corresponding to NiO, pointing to the corresponding cell planes (1 1 1), (2 2 0), and (2 2 2), which is consistent with the NiO measured using XRD. It indicates that nanorods with a diameter of about 10 nm are formed on the nanosheet structure after electrophoretic deposition, which is also the main reason for improving specific capacitance. During electrophoretic deposition, the following reactions mainly occurred:H_2_O + Ni^+^ → H_2_ + NiO + e^+^(5)

In order to further determine the elemental composition of nanorods, STEM-EDX analysis was carried out. As shown in Figure 6c–f, the main elements of nanorods are Ni and O elements, which are evenly distributed, which proves that the main component of the surface rod structure is NiO. These results are consistent with XRD and XPS analysis [41]. Due to the excellent theoretical specific capacitance and thermal stability of NiO, the electrochemical properties of the prepared electrode materials can be greatly improved by attaching NiO to NiCo(OH)_4_ as nanorod-like structure. In addition, the composite electrode material NiCo(OH)_4_-NiO can contact more point ions in the electrolyte solution, which effectively reduces the contact resistance and improves the electron transport rate. The nanoscale petal spherical electrode material composed of two Faraday REDOX materials plays an important role in improving the electrochemical performance of pseudocapacitors. Figure 6g,h show the physical images of nickel foam before and after electrophoretic deposition. It can be seen that the foamed nickel changes from brown to brownish green after electrophoretic deposition. During electrophoretic deposition, a large number of bubbles was observed near the electrode, which is the result of H_2_ overflow during the electrophoretic deposition.

### 3.3. Electrochemical Characterization

Figure 7a shows a comparison of charge–discharge diagrams of the prepared NiCo(OH)_4_-NiO-70, NiCo(OH)_4_-NiO-90, NiCo(OH)_4_-NiO-100 and NiCo(OH)_4_-NiO-110 electrodes at 1 A g^−1^ [39,40]. NiCo(OH)_4_-NiO-90 has the longest discharge time compared with NiCo(OH)_4_-NiO-70, NiCo(OH)_4_-NiO-100, and NiCo(OH)_4_-NiO-110 electrodes. This indicates that with the increase in the reaction temperature, the pore spacing of nano-scale porous electrode material increases, and ions are more easily diffused. However, with the increase in temperature, the bond of the electrode material is broken, and the pore collapses. It cannot provide more active sites for ions, affecting the electrochemical performance [25]. The specific capacitances of NiCo(OH)_4_-NiO-70, NiCo(OH)_4_-NiO-90, NiCo(OH)_4_-NiO-100, and NiCo(OH)_4_-NiO-110 curves calculated at 1 mV s^−1^ are 757 F g^−1^, 2129 F g^−1^, 1767 F g^−1^, and 1616 F g^−^^1^, respectively. The electrochemical performance of NiCo(OH)_4_-NiO-90 electrode material is significantly higher than that of NiO@Ni(OH)_2_ electrode material (1205 F g^−1^) prepared by Mozaffari et al. [27]. This is because of the existence of Co element that has better conductivity, and the electrode material NiCo(OH)_4_-NiO-90, which can provide more chemical valence. Both of these reasons can effectively improve the electrochemical performance. Compared with other composite materials containing Mo, S, and CNT, NiCo(OH)_4_-NiO-90 has larger electrochemical properties and a simple preparation process without pollution [25,33,37]. As shown in Figure 7b, the calculated specific capacitors of NiCo(OH)_4_-NiO-90 at different current densities of 1, 2, 5, 10, and 20 A g^−1^ are 2129 F g^−1^, 1695 F g^−1^, 1366 F g^−1^, 1156 F g^−1^, and 922 F g^−^^1^, respectively, indicating that NiCo(OH)_4_-NiO-90 has excellent electrochemical properties [42]. Figure 7b shows the charge–discharge curves of NiCo(OH)_4_-NiO-90 at different current densities. The measured specific capacitance reaches the maximum of 2129 F g^−1^ at 1 A g^−1^ current density. When the current density increases, the ion concentration adsorbed on the electrode interface in the electrolytic cell decreases rapidly. At the same time, a high excitation voltage is needed to maintain high current density. Therefore, without increasing the amount of interfacial charge, the capacitance measured at high current density is much smaller than that measured at low current density [15,43,44].

Cyclic voltammetry (CV) was performed for NiCo(OH)_4_-NiO-70, NiCo(OH)_4_-NiO-90, NiCo(OH)_4_-NiO-100 and NiCo(OH)_4_-NiO-110 in a potential window of 0 to 0.6 V at 1 mV s^−1^. The CV curve is shown in Figure 7c. The recording curves of the three electrodes reveal nonlinear properties close to the REDOX properties of the battery. A specific pair of REDOX peaks is observed with a sharp [34,45], upright oxidation peak, which is typical for supercapacitors. As shown in Figure 7c, the unique oxidation peak is caused by the REDOX reaction formed by the gain and loss of NiO electrons in the charging and discharging process, which is similar to the REDOX reaction mechanism of supercapacitors previously reported [27]. In addition, the reduction in -OH^−^ results in a large area of sharp reduction peaks. However, of the three electrodes, NiCo(OH)_4_-NiO-90 shows the best performance (shown in Figure 7c), with high current density and the largest integral area of the REDOX reaction peak. This is most similar to the NiCo(OH)_4_-NiO-90 performance measured in Figure 7a.

Figure 7e shows the comparison of the measured specific capacitance of the prepared electrode materials NiCo(OH)_4_-NiO-70, NiCo(OH)_4_-NiO-90, NiCo(OH)_4_-NiO-100 and NiCo(OH)_4_-NiO-110 under different current densities. It is obvious that NiCo(OH)_4_-NiO-90 has the largest specific capacitance, for the reasons explained in the previous section.

By testing the resistance of four different electrode materials, the impedance diagram as shown in Figure 7f is obtained. The illustration shows that the equivalent circuit consists of internal solution resistance Rs, Faraday charge transfer resistance Rct, and Warburg impedance W. The straight-line part of the curve represents the Warburg impedance of the material. In other words, the higher the slope of the line is, the lower the ionic diffusion resistance of the material is. The ion diffusion impedance of NiCo(OH)_4_-NiO-90 electrode material is the smallest. Combined with the SEM figure, it can be seen that the holes between the nanosheet structures on the surface of NiCo(OH)_4_-NiO-90 electrode material are larger, which is convenient for the transfer of charge and ions with points and can provide a higher-rate performance [46]. At the same time, NiCo(OH)_4_-NiO-90 electrode material has the largest specific surface area, which is beneficial to the sufficient contact between electrolyte and electrode, and has a faster redox reaction rate, thus reducing the charge transfer resistance.

Figure 7g shows the specific capacitance retention of NiCo(OH)_4_-NiO-90 electrode material after 1000 charge–discharge cycles at the current density of 1 A g^−1^. After 1000 cycles, the specific capacitance still has 85% of the initial value. This is because in the process of charge–discharge, irreversible REDOX reaction occurs, the continuous consumption of NiO and -OH^−^ in the reaction process leads to the reduction of active substances, which is also the characteristic of pseudocapacitor [47].

Through the electrochemical performance test, it can be confirmed that the electrochemical performance of the composites prepared in this work is better than other similar composites reported in previous literature. (Table 1)

## 4. Conclusions

In this work, NiCo(OH)_4_-NiO composite electrode materials with spherical flower structure were prepared using a hydrothermal method combined with electrophoretic deposition. NiO is deposited on the surface of NiCo(OH)_4_ petals as nanorods, forming a porous structure with large specific area. NiCo(OH)_4_ acts synergically with NiO deposited on the surface and has a large specific surface area, thus improving the specific capacitance. Meanwhile, the porous structure is conducive to charge transfer and improves the conductivity. Through the comparison of different hydrothermal reaction temperatures, it is concluded that when the hydrothermal reaction temperature is 90 °C, the measured specific capacitance is the largest. The measured specific capacitance reaches the maximum of 2129 F g^−1^ at 1 A g^−1^ current density. This is because, with the increase in the reaction temperature, the crystal growth rate in the solution is faster, and it is easier to form petal spherules. When the temperature is higher than 90 °C, the flake structure appears to be a stacking agglomeration phenomenon, and the spacing between the nanosheets becomes smaller, leading to reduction of the specific surface area. From 90 °C to 110 °C, the distance between the nanosheets becomes smaller and smaller, and the measured specific capacitance also shows a decreasing trend. After 1000 charge–discharge cycles, NiCo(OH)_4_-90 can still maintain 85% specific capacitance, which is due to the stability of the nanosheet structure formed, so that the material has good cycling performance. Overall, although there are many studies on composite electrode materials, there are few studies on the performance of the prepared materials with specific experimental parameters. In this paper, a simple preparation process was used to obtain the electrode material with the best electrochemical performance by controlling the hydrothermal reaction temperature. It indicates that the electrochemical performance can be optimized by adjusting the microstructure of electrode materials, which provides guidance for the performance optimization of composite electrode materials.

## Figures and Tables

**Figure 1 nanomaterials-12-02276-f001:**
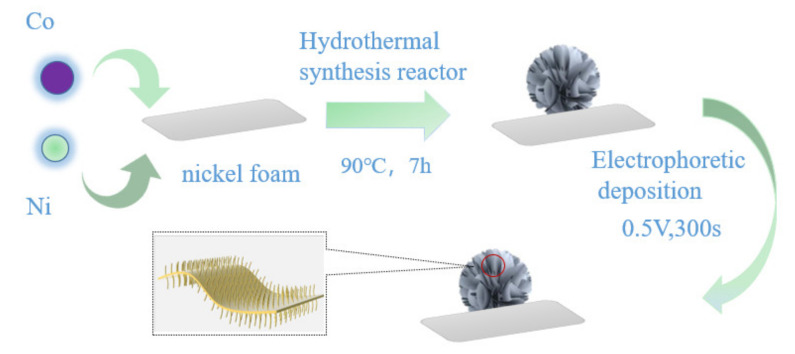
NiCo(OH)_4_-NiO preparation diagram.

**Figure 2 nanomaterials-12-02276-f002:**
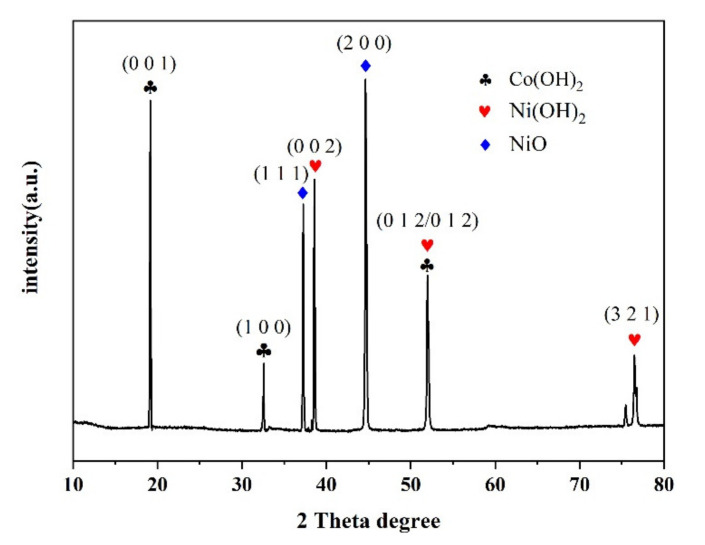
XRD patterns of NiCo(OH)_4_-NiO.

**Figure 3 nanomaterials-12-02276-f003:**
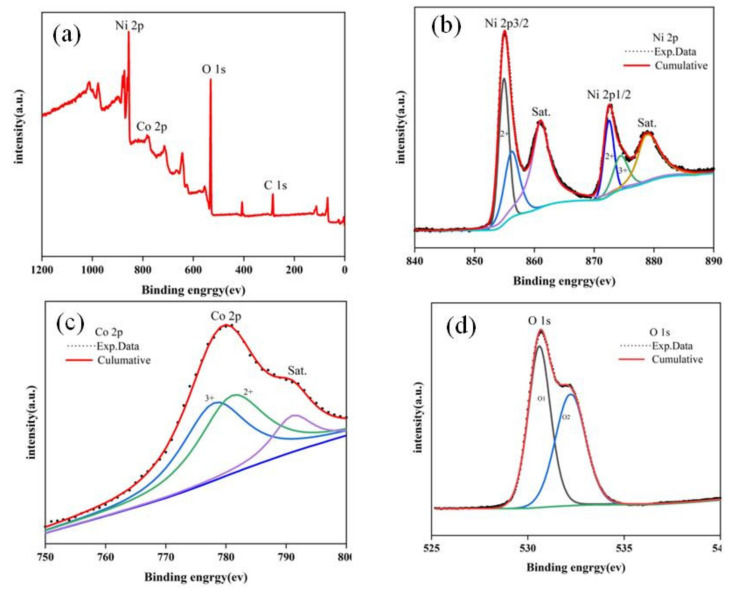
XPS spectra of NiCo (OH)_4_-NiO nanocomposite: (**a**) survey spectrum, (**b**) Ni 2p, (**c**) Co 2p, (**d**) O 1s.

**Figure 4 nanomaterials-12-02276-f004:**
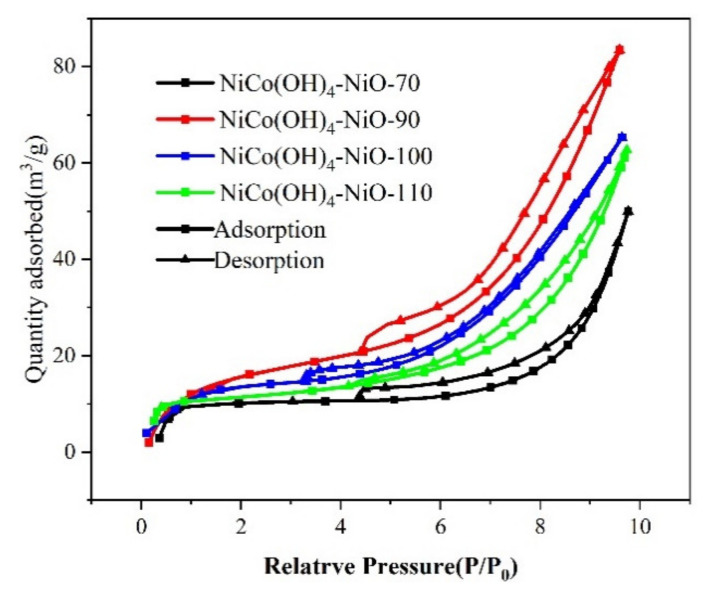
N_2_ adsorption–desorption isotherms of NiCo(OH)_4_-NiO electrode materials.

**Figure 5 nanomaterials-12-02276-f005:**
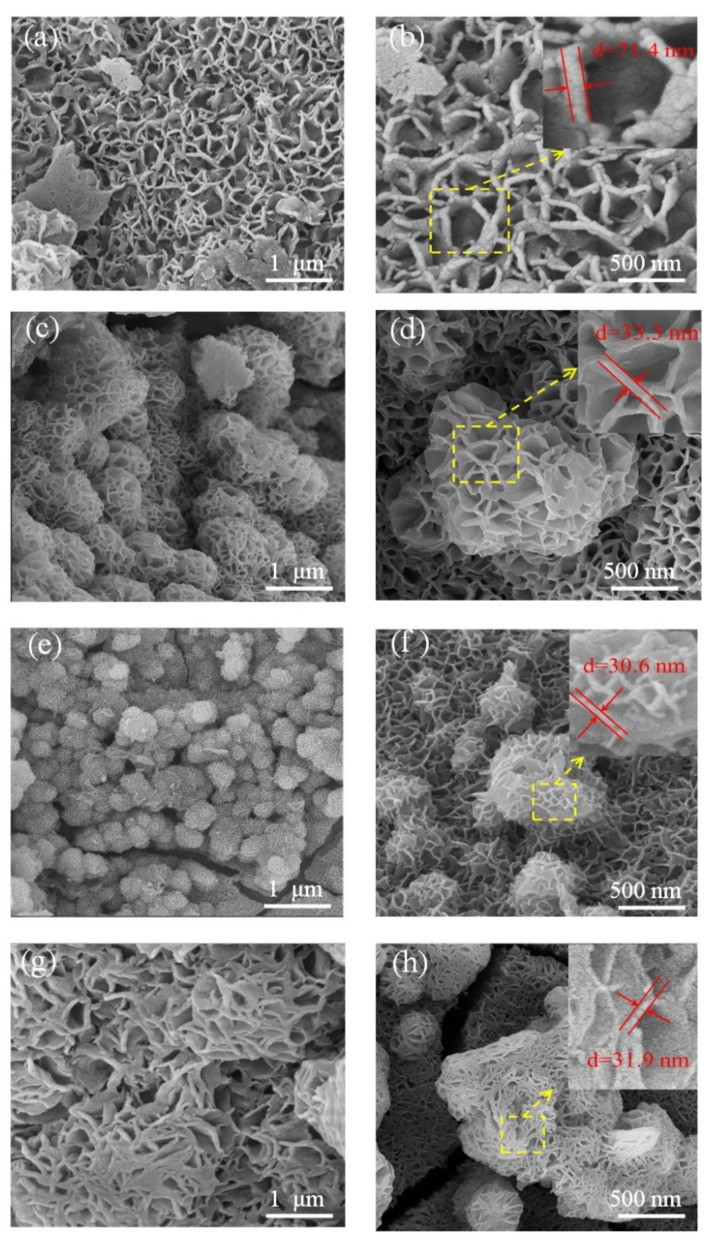
SEM images of as-prepared samples: (**a**,**b**) NiCo(OH)_4_-NiO-70 at different separation rates, (**c**,**d**) NiCo(OH)_4_-NiO-90 at different separation rates, (**e**,**f**) NiCo(OH)_4_-NiO-100 at different separation rates, (**g**,**h**) NiCo(OH)_4_-NiO-110 at different separation rates.

**Figure 6 nanomaterials-12-02276-f006:**
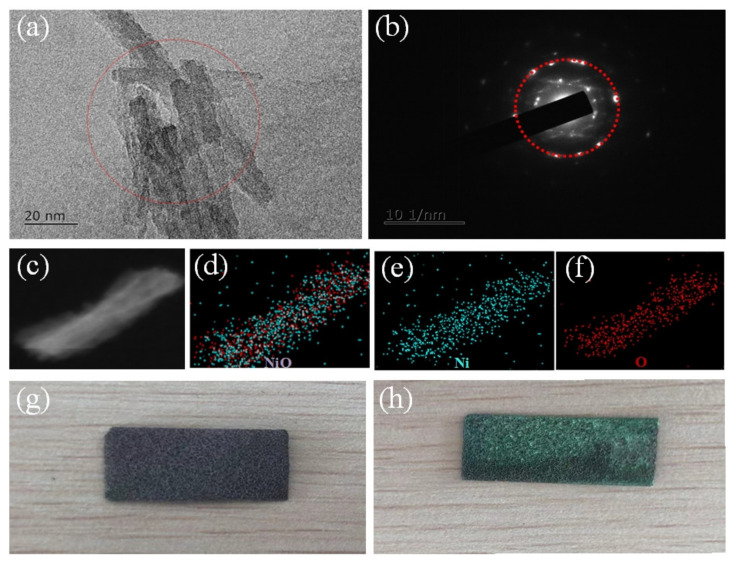
(**a**) TEM of NiO, (**b**) the corresponding SAED pattern, (**c**–**f**), STEM-EDX elemental mappings of NiO, (**g**,**h**) Photos of foamed nickel before and after electrophoretic deposition.

**Figure 7 nanomaterials-12-02276-f007:**
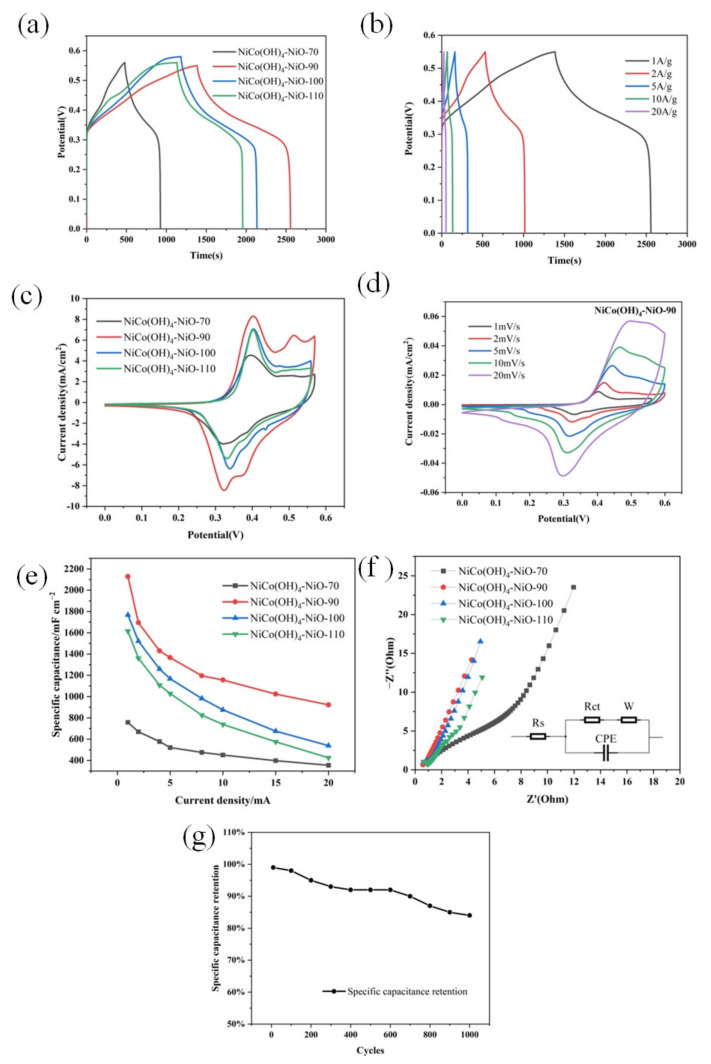
(**a**) Charge–discharge curves of electrode materials prepared at different hydrothermal reaction temperatures at 1 A g^−1^, (**b**) charge–discharge curves of NiCo(OH)_4_-NiO-90 electrode materials at different current densities, (**c**) REDOX reaction of NiCo(OH)_4_-NiO-90 electrode materials at different scanning rates, (**d**) REDOX reaction of electrode materials prepared at different hydrothermal reaction temperatures at 1 mV s^−1^, (**e**) comparison of specific capacitance of electrode materials prepared at different hydrothermal reaction temperatures, (**f**) impedance comparison of electrode materials prepared at different hydrothermal reaction temperatures; the inset shows an equivalent circuit diagram, (**g**) specific capacitance retention of NiCo(OH)_4_-NiO-90 after 1000 charge and discharge cycles.

**Table 1 nanomaterials-12-02276-t001:** The electrochemical performance between NiCo(OH)_4_-NiO and other electrode materials reported in previous literature.

Material	Specific Capacitance (F g^−1^)	Cycle Performance	References
NiO@Ni(OH)_2_	1205	73%, (5000)	[27]
NiO/Ni@C	1610	70%, (1000)	[21]
NiCo-LDH@MoO_3_/NF	1904	86%, (10,000)	[25]
NiCo(NA)-LDH	1907	66%, (1000)	[48]
Ni-Co@Co-MOF/NF	1997	80%, (10,000)	[49]
NiCo(OH)_4_-NiO	2129	85%, (1000)	This work

## Data Availability

The raw/processed data required to reproduce these findings cannot be shared at this time due to technical or time limitations.

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
