# Peer review of "Effect of Hydrothermal Method Temperature on the Spherical Flowerlike Nanostructures NiCo(OH)_4_-NiO"

_nanomaterials, 2022, doi:10.3390/nano12132276_

Round 1

Reviewer 1 Report

The title of the work is "Preparation and analysis of high-performance spherical flowerlike nanostructures NiCo(OH)4-NiO". The preparation of the NiCo(OH)4-NiO is well reported and the characterization part is also well discussed. I have minor comments before accept the manuscript. 

Authors mentioned high-performance of the material, how it is proven. I suggest authors to compare the previously reported similar  type of materials  with the present data.

Author Response

Dear editor

      Thank you for your valuable comments,  I have made corresponding changes according to your comments.  The specific content is attached.

Reviewer 2 Report

Comments: This paper reports the preparation and analysis of high-performance spherical flower-like nanostructures NiCo(OH)4-NiO. Authors should address the following comments for its acceptance.

1.    How is the synthesis method different or better than those reported earlier? Author should highlight this in the introduction part.

2.    Title needs to be clear

3.    Author needs to extend the charge-discharge cycles up to 5000 or 10000 to show the superiority of the synthesized materials.

4.    STEM-EDX elemental mappings of NiO are not clear.

5.    Author needs to fit the EIS Nyquist plots with a suitable circuit diagram and a corresponding discussion should be included for the understanding of electrode and electrolyte interaction during the electrochemical reaction.

6.    What about the reusability of prepared materials towards supercapacitors.

7.    In order to show the superiority of the current materials, comparisons over the other related materials reported in the past works of literature are necessary. Supercapacitor performances of the current materials have to be compared with those of the other materials and reasons for performance improvements have to be discussed.

8.    Conclusion should be improved with novelty and be the importance of the investigation in the present work.

Author Response

Dear editor:

Thank you for your valuable comments. I have made corresponding changes according to your comments. The specific content is attached.

Reviewer 3 Report

This paper titled “Preparation and analysis of high-performance spherical flowerlike nanostructures NiCo(OH)4-NiO” addresses the hydrothermal synthesis of  NiCo(OH)4 on Ni foam at different conditions with further deposition of NiO nanoroads on the surface of obtained material. The composites were investigated as electrodes of capacitor using 1M KOH solution as electrolyte. A specific capacitance of 2129 F g-1 at 1 A g-1 current density is reported. Through the comparison of different hydrothermal reaction temperatures, the largest specific capacitance achieved for composite obtained at 90C.

Conclusions not sufficiently supported by experimental data. 

1.       Four composites were synthesised by hydrothermaly at differnet reaction temperatures. And authors note that increase of temperature leads the faster growth rate of nanosheets of NiCo(OH)4. It should be better to provide XRD data of all 4 materials and discuss average crystallite size.

2.       While describing the materials, the authors repeatedly underline the change in the specific surface area for materials obtained at different temperatures. However, there are no experimental data to support this fact. Apparently, this parameter is estimated from electron microscopy images. Please provide BET data.

3.       There are no any conformation of thickness of electrode material changing: “When the hydrothermal reaction temperature increased from 70 ℃ to 90 ℃, the thickness of the electrode material microstructure nanosheets becomed thinner

4.       The text contains typos and inaccuracies: “The Figure 6a shows a comparison of charge-discharge diagrams of the prepared NiCo(OH)4-NiO-80”;

5.       The language should be further polished. Especially, grammatical issues that are frequently seen throughout this paper

Author Response

(The authors gave the same response as above.)

Round 2

Reviewer 2 Report

The revised version of the manuscript may acceptable to the journal standard.

Reviewer 3 Report

Paper titeld "Preparation and analysis of high-performance spherical flowerlike nanostructures NiCo(OH)4-NiO" may be accepted for publication in present form.